# Bootstrapping Semantic Segmentation with Regional Contrast

**Shikun Liu[1], Shuaifeng Zhi[1], Edward Johns[2], and Andrew J. Davison[1]**
[1]Dyson Robotics Lab, Imperial College London
[2]Robot Learning Lab, Imperial College London
`shikun.liu17@imperial.ac.uk`

## Abstract

We present ReCo, a contrastive learning framework designed at a regional level to assist learning in semantic segmentation. ReCo performs pixel-level contrastive learning on a sparse set of hard negative pixels, with minimal additional memory footprint. ReCo is easy to implement, being built on top of off-the-shelf segmentation networks, and consistently improves performance, achieving more accurate segmentation boundaries and faster convergence. The strongest effect is in semi-supervised learning with very few labels. With ReCo, we achieve high quality semantic segmentation model, requiring only 5 examples of each semantic class.

## 1 Introduction

Semantic segmentation is an essential part of applications such as scene understanding and autonomous driving, whose goal is to assign a semantic label to each pixel in an image. Significant progress has been achieved by use of large datasets with high quality human annotations. However, labelling images with pixel-level accuracy is time consuming and expensive; for example, labelling a single image in CityScapes can take more than 90 minutes (Cordts et al., 2016). When deploying semantic segmentation models in practical applications where only limited labelled data are available, high quality ground-truth annotation is a significant bottleneck.

To reduce the need for labelled data, there is a recent surge of interest in leveraging unlabelled data for semi-supervised learning. Previous methods include improving segmentation models via adversarial learning (Hung et al., 2019; Mittal et al., 2019) and self-training (Zou et al., 2019; 2018; Zhu et al., 2020). Others focus on designing advanced data augmentation strategies to generate pseudo image-annotation pairs from unlabelled images (Olsson et al., 2021; French et al., 2020).

In both semi-supervised and supervised learning, a segmentation model often predicts smooth label maps, because neighbouring pixels are usually of the same class, and rarer high-frequency regions are typically only found in object boundaries. This learning bias produces blurry contours and regularly mis-labels rare objects. After carefully examining the label predictions, we further observe that wrongly labelled pixels are typically confused with very few other classes; *e.g.* a pixel labelled as `rider` has a much higher chance of being wrongly classified as `person`, compared to `train` or `bus`. By understanding this class structure, learning can be *actively* focused on the challenging pixels to improve overall segmentation quality.

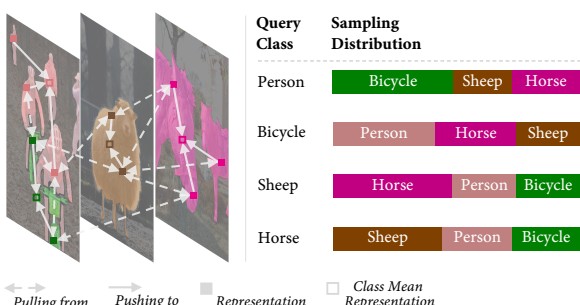

Figure 1: ReCo pushes representations within a class closer to the class mean representation, whilst simultaneously pushing these representations away from negative representations sampled in different classes. The sampling distribution from negative classes is adaptive to each query class.

Here we propose ReCo, a contrastive learning framework designed at a regional level. Specifically, ReCo is a new loss function which helps semantic segmentation not only to learn from local context

(neighbouring pixels), but also from global semantic class relationships across the entire dataset. ReCo performs contrastive learning on a pixel-level dense representation, as visualised in Fig. 1. For each semantic class in a mini-batch, ReCo samples a set of pixel-level representations (queries), and encourages them to be close to the class mean representation (positive keys), and simultaneously pushes them away from representations sampled from other classes (negative keys).

For pixel-level contrastive learning with high-resolution images, it is impractical to sample all pixels. In ReCo, we actively sample a sparse set of queries and keys, consisting of *less than 5%* of all available pixels. We sample negative keys from a learned distribution based on the relative distance between the mean representation of each negative key and the query class. This distribution can be interpreted as a pairwise semantic class relationship, dynamically updated during training. We sample queries for those having a low prediction confidence. Active sampling helps ReCo to rapidly focus on the most confusing pixels for each semantic class, and requires minimal additional memory.

ReCo enables a high-accuracy segmentation model to be trained with very few human annotations. We evaluate ReCo in a semi-supervised setting, with two different modes: i) *Partial Dataset Full Labels* — a sparse subset of training images, where each image has full ground-truth labels, and the remaining images are unlabelled; ii) *Partial Labels Full Dataset* — all images have some labels, but covering only a sparse subset of pixels within each image. In both settings, we show that ReCo can consistently improve performance across all methods and datasets.

## 2 RELATED WORK

**Semantic Segmentation** One recent direction is in designing more effective deep convolutional neural networks. Fully convolutional networks (FCNs) (Long et al., 2015) are the foundation of modern segmentation network design. They were later improved with dilated/atrous convolutions with larger receptive fields, capturing more long range information (Chen et al., 2017; 2018). Alternative approaches include encoder-decoder architectures (Ronneberger et al., 2015; Kirillov et al., 2019), sometimes using skip connections (Ronneberger et al., 2015) to refine filtered details.

A parallel direction is to improve optimisation strategies, by designing loss functions that better respect class imbalance (Lin et al., 2017) or using rendering strategy to refine uncertain pixels from high-frequency regions improving the label quality (Kirillov et al., 2020). ReCo is built upon this line of research, to improve segmentation by providing additional supervision on hard pixels.

**Semi-supervised Classification and Segmentation** The goal of semi-supervised learning is to improve model performance by taking advantage of a large amount of unlabelled data during training. Here consistency regularisation and entropy minimisation are two common strategies. The intuition is that the network's output should be invariant to data perturbation and geometric transformation. Based on these strategies, many semi-supervised methods have been developed for image classification (Sohn et al., 2020; Tarvainen & Valpola, 2017; Berthelot et al., 2019; Kuo et al., 2020).

However, for segmentation, generating effective pseudo-labels and well-designed data augmentation are non-trivial. Some solutions improved the quality of pseudo-labelling, using adversarial learning (Hung et al., 2019; Mittal et al., 2019) or enforcing consistency from different augmented images (French et al., 2020; Olsson et al., 2021). In this work, we show that we can improve the performance of current semi-supervised segmentation methods by jointly training with a suitable auxiliary task.

**Contrastive Learning** Contrastive learning learns a similarity function to bring views of the same data closer in representation space, whilst pushing views of different data apart. Most recent contrastive frameworks learn similarity scores based on *global representations* of the views, parameterising data with a single vector (He et al., 2020; Chen et al., 2020; Khosla et al., 2020). *Dense representations*, on the other hand, rely on pixel-level representations and naturally provide additional supervision, capturing fine-grained pixel correspondence. Contrastive pre-training based on dense representations has recently been explored, and shows better performance in dense prediction tasks, such as object detection and keypoint detection (Wang et al., 2021b; O. Pinheiro et al., 2020).

**Contrastive Learning for Semantic Segmentation** Contrastive learning has been recently studied to improve semantic segmentation, with a number of different design strategies. Zhang et al. (2021) and Zhao et al. (2021) both perform contrastive learning via pre-training, based on the generated auxiliary labels and ground-truth labels respectively, but at the cost of huge memory consump-

tion. In contrast, ours performs contrastive learning whilst requiring much less memory, via active sampling. In concurrent work, (Wang et al., 2021a; Alonso et al., 2021) also perform contrastive learning with active sampling. However, whilst both these methods are applied to a stored feature bank, ours focuses on sampling features on-the-fly. Active sampling in Alonso et al. (2021) is further based on learnable, class-specific attention modules, whilst ours only samples features based on relation graphs and prediction confidence, without introducing any additional computation overhead, which results in a simpler and much more memory-efficient implementation.

## 3 ReCo – Regional Contrast

### 3.1 Pixel-Level Contrastive Learning

Let $(X, Y)$ be a training dataset with training images $x \in X$ and their corresponding $C$-class pixel-level segmentation labels $y \in Y$, where $y$ can be either provided in the original dataset, or generated automatically as pseudo-labels. A segmentation network $f$ is then optimised to learn a mapping $f_\theta : X \mapsto Y$, parameterised by network parameters $\theta$. This segmentation network $f$ can be decomposed into two parts: an encoder network: $\phi : X \mapsto Z$, and a decoder classification head $\psi_c : Z \mapsto Y$. To perform pixel-level contrastive learning, we additionally attach a decoder representation head $\psi_r$ on top of the encoder network $\phi$, parallel to the classification head, mapping the encoded feature into a higher $m$-dimensional dense representation with the same spatial resolution as the input image: $\psi_r : Z \mapsto R, R \in \mathbb{R}^m$. This representation head is only applied during training to guide the classifier using the ReCo loss as an auxiliary task, and is removed during inference.

A pixel-level contrastive loss is a function which encourages queries $r_q$ to be similar to the positive key $r_k^+$, and dissimilar to the negative keys $r_k^-$. All queries and keys are sampled from the decoder representation head: $r_q, r_k^{+,-} \in R$. In ReCo, we use a pixel-level contrastive loss across all available semantic classes in each mini-batch, with the distance between keys and queries measured by their normalised dot product. The general formation of the ReCo loss $L_{\texttt{reco}}$ is then defined as:

$$L_{\texttt{reco}} = \sum_{c \in \mathcal{C}} \sum_{r_q \sim \mathcal{R}_q^c} - \log \frac{\exp(r_q \cdot r_k^{c,+}/\tau)}{\exp(r_q \cdot r_k^{c,+}/\tau) + \sum_{r_k^- \sim \mathcal{R}_k^c} \exp(r_q \cdot r_k^-/\tau)}, \qquad (1)$$

for which $\mathcal{C}$ is a set containing all available classes in the current mini-batch, $\tau$ is the temperature control of the softness of the distribution, $\mathcal{R}_q^c$ represents a query set containing all representations whose labels belong to class $c$, $\mathcal{R}_k^c$ represents a negative key set containing all representations whose labels do not belong to class $c$, and $r_k^{c,+}$ represents the positive key which is the mean representation of class $c$. Suppose $\mathcal{P}$ is a set containing all pixel coordinates with the same resolution as $R$, these queries and keys are then defined as:

$$\mathcal{R}_q^c = \bigcup_{[u,v] \in \mathcal{P}} \mathbb{1}(y_{[u,v]} = c) r_{[u,v]}, \ \mathcal{R}_k^c = \bigcup_{[u,v] \in \mathcal{P}} \mathbb{1}(y_{[u,v]} \neq c) r_{[u,v]}, \ r_k^{c,+} = \frac{1}{|\mathcal{R}_q^c|} \sum_{r_q \in \mathcal{R}_q^c} r_q. \quad (2)$$

### 3.2 Active Hard Sampling on Queries and Keys

Contrastive learning on all pixels in high-resolution images would be computationally expensive. Here, we introduce active hard sampling strategies to optimise only a sparse set of queries and keys.

**Active Key Sampling**  When classifying a pixel, a semantic network might be uncertain only over a very small number of candidates, among all available classes. The uncertainty from these candidates typically comes from a close spatial (*e.g.* `rider` and `bicycle`) or semantic (*e.g.* `horse` and `cow`) relationship. To reduce this uncertainty, we propose to sample negative keys non-uniformly, based on the relative distance between each negative key class and the query class. This involves building a pair-wise class relationship graph $G$, with $G \in \mathbb{R}^{|\mathcal{C}| \times |\mathcal{C}|}$, computed and dynamically updated for each mini-batch. This pair-wise relationship is measured by the normalised dot product between the mean representation from a pair of two classes and is defined as:

$$G[p, q] = \left( r_k^{p,+} \cdot r_k^{q,+} \right), \quad \forall p, q \in \mathcal{C}, \text{ and } p \neq q. \qquad (3)$$

We further apply SoftMax to normalise these pair-wise relationships among all negative classes $j$ for each query class $c$, which produces a probabilistic distribution:

$\exp(G[c, i])/\sum_{j \in \mathcal{C}, j \neq c} \exp(G[c, j])$. We sample negative keys for each class $i$ based on this distribution, to learn the corresponding query class $c$. This procedure allocates more samples to hard, confusing classes chosen specifically for each query class, helping the segmentation network to learn a more accurate decision boundary.

**Active Query Sampling**  Due to the natural class imbalance in semantic segmentation, it is easy to over-fit on common classes, such as the `road` and `building` classes in the CityScapes dataset, or the `background` class in the Pascal VOC dataset. These common classes contribute to the majority of pixel space in training images, and so randomly sampling queries will under-sample rare classes and provide minimal supervision to these classes.

Therefore, we instead sample hard queries — for those whose corresponding pixel prediction confidence is below a defined threshold. Accordingly, ReCo's loss would then guide the segmentation network by providing appropriate supervision on these less certain pixels. The easy and hard queries are defined as follows, and visualised in Fig. 2,

$$\mathcal{R}_q^{c,\,easy} = \bigcup_{r_q \in \mathcal{R}_q^c} \mathbb{1}(\hat{y}_q > \delta_s)r_q, \quad \mathcal{R}_q^{c,\,hard} = \bigcup_{r_q \in \mathcal{R}_q^c} \mathbb{1}(\hat{y}_q \leq \delta_s)r_q, \tag{4}$$

where $\hat{y}_q$ is the predicted confidence of label $c$ after the SoftMax operation corresponding to the same pixel location as $r_q$, and $\delta_s$ is the user-defined confidence threshold.

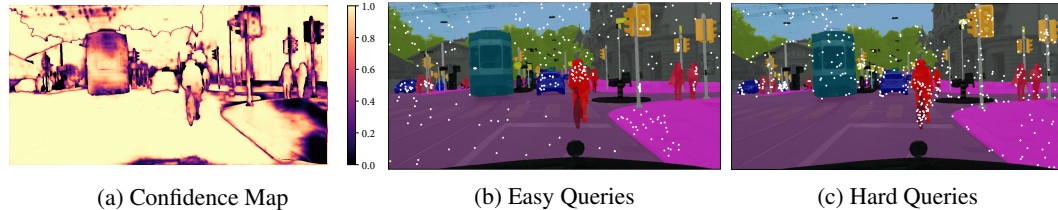

(a) Confidence Map          (b) Easy Queries          (c) Hard Queries

Figure 2: Easy and hard queries (shown in white) determined from the predicted confidence map in the Cityscapes dataset. Here we set the confidence threshold $\delta_s = 0.97$.

### 3.3  SEMI-SUPERVISED SEMANTIC SEGMENTATION WITH RECO

ReCo can easily be added to modern semi-supervised segmentation methods without changing the training pipeline, with *no additional cost* at inference time. To incorporate ReCo, we simply add an additional representation head $\psi_r$ as described in Section 3.1, and apply the ReCo loss (in Eq. 1) to this representation using the sampling strategy introduced in Section 3.2.

We apply the Mean Teacher framework (Tarvainen & Valpola, 2017) following prior state-of-the-art semi-supervised segmentation methods (Olsson et al., 2021; Mittal et al., 2019). Instead of using the original segmentation network $f_\theta$ (which we call the student model), we instead use $f_{\theta'}$ (which we call the teacher model) to generate pseudo-labels from unlabelled images, where $\theta'$ is a moving average of the previous state of $\theta$ during training optimisation: $\theta_t' = \lambda \theta_{t-1}' + (1-\lambda)\theta_t$, with a decay parameter $\lambda = 0.99$. This teacher model can be treated as a temporal ensemble of student models across training time $t$, resulting in more stable predictions for unlabelled images. The student model $f_\theta$ is then used to train on the augmented unlabelled images, with pseudo-labels as the ground-truths.

For all pixels with defined ground-truth labels, we apply the ReCo loss on dense representations corresponding to all valid pixels. For all pixels without such labels, we only sample pixels whose predicted pseudo-label confidence is greater than a threshold $\delta_w$. This avoids sampling pixels which are likely to have incorrect pseudo-labels.

We apply the ReCo loss to a combined set of labelled and unlabelled pixels. The overall training loss for semi-supervised segmentation is then the linear combination of supervised cross-entropy loss (on ground-truth labels), unsupervised cross-entropy loss (on pseudo-labels) and ReCo loss:

$$L_{total} = L_{supervised} + \eta \cdot L_{unsupervised} + L_{reco}, \tag{5}$$

where $\eta$ is defined as the percentage of pixels whose predicted confidence are greater than $\delta_s$, a scalar re-weighting the contribution for unsupervised loss, following prior methods (Olsson et al., 2021;

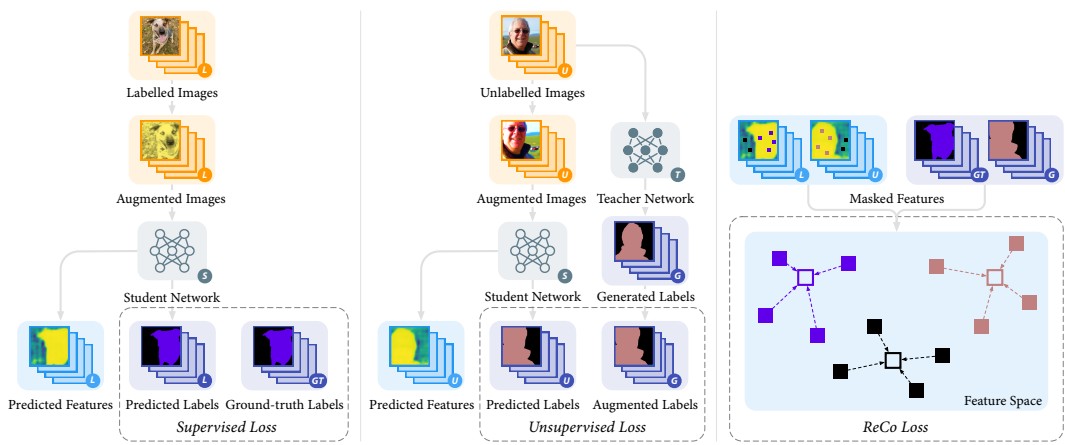

Figure 3: Visualisation of the ReCo framework applied to semi-supervised segmentation and trained with three losses. A supervised loss is computed based on labelled data with ground-truth annotations. An unsupervised loss is computed for unlabelled data with generated pseudo-labels. And finally a ReCo loss is computed based on pixel-level dense representation predicted from both labelled and unlabelled images.

Mittal et al., 2019). This makes sure the segmentation network would not be dominated by gradients produced by uncertain pseudo-labels, which typically occur during the early stage of training. Fig. 3 shows a visualisation of the ReCo framework for semi-supervised segmentation.

## 4 EXPERIMENTS

**Semi-Supervised Segmentation Benchmark Redesign** We propose two modes of semi-supervised segmentation tasks, aiming at two different applications.

i) *Partial Dataset Full Labels*: A small subset of the images is trained with complete ground-truth labels for each image, whilst the remaining training images are unlabelled. This is the de-facto standard of evaluating semi-supervised segmentation in prior works.

ii) *Partial Labels Full Dataset*: All images are trained with partial labels, but only a small percentage of labels are provided for each class in each training image. We create the dataset by first randomly sampling a pixel for each class, and then continuously apply a $[5 \times 5]$ square kernel for dilation until we meet the percentage criteria.

The Partial Dataset Full Label setting evaluates the ability to generalise semantic classes given a few examples with perfect boundary information. The Partial Label Full Dataset evaluates learning semantic class completion given many examples with no or minimal boundary information.

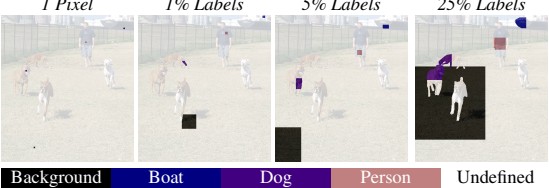

Figure 4: Example of training labels for Pascal VOC dataset in Partial Labels Full Dataset setting. (1 Pixel is zoomed 5 times for better visualisation.)

**Datasets** We experiment on segmentation datasets: Cityscapes (Cordts et al., 2016) and Pascal VOC 2012 (Everingham et al., 2015) in both partial and full label setting. We also evaluate on a more difficult indoor scene dataset SUN RGB-D (Song et al., 2015) in the full label setting only, mainly due to the low quality annotations making it difficult for fair evaluation in the partial label setting. An example of the partially labelled Pascal VOC is shown in Fig. 4.

**Strong Baselines** Prior semi-supervised segmentation methods are typically designed with different backbone architectures, and trained with different strategies, which makes it difficult to compare them fairly. In this work, we standardise the baselines and implement four strong semi-supervised segmentation methods ourselves: *S4GAN* (Mittal et al., 2019): an adversarial learning based semi-supervised method; *CutOut* (French et al., 2020), *CutMix* (French et al., 2020), *ClassMix* (Olsson

et al., 2021): three image augmentation strategies designed specifically for semi-supervised segmentation. Our implementations for all baselines obtain performance on par with, and most of the time surpassing, the performance reported in each original publication, giving us a set of strong baselines. All baselines and our method were implemented on DeepLabV3+ (Chen et al., 2018) with ResNet-101 backbone (He et al., 2016), and all with the same optimisation strategies. Detailed hyper-parameters used for each dataset are provided in the Appendix A.

## 4.1 RESULTS ON PASCAL VOC, CITYSCAPES, SUN RGB-D (FULL LABELS)

First, we compared our results to semi-supervised baselines in a full label setting. We applied ReCo on top of ClassMix, which consistently outperformed other semi-supervised baselines.

Table 1 shows the mean IoU validation performance on three datasets over three individual runs (different labelled and unlabelled data splits). The number of labelled images shown in the three columns for each dataset, are chosen such that the least-appeared classes have appeared in 5, 15 and 50 images respectively. In the fewest-label setting for each dataset, applying ReCo with ClassMix can improve results by a significant margin, with up to $1.5 - 4.5\%$ absolute improvement.

| Method | Pascal VOC | | | CityScapes | | | SUN RGB-D | | |
|---|---|---|---|---|---|---|---|---|---|
| | 60 labels | 200 labels | 600 labels | 20 labels | 50 labels | 150 labels | 50 labels | 150 labels | 500 labels |
| Supervised | 37.79 | 53.87 | 64.04 | 38.12 | 45.42 | 54.93 | 19.79 | 28.78 | 37.73 |
| S4GAN (Mittal et al., 2019) | 47.95 | 61.25 | 66.21 | 37.65 | 47.08 | 56.46 | 20.53 | 29.79 | 38.08 |
| CutOut (French et al., 2020) | 52.96 | 63.57 | 69.85 | 42.52 | 50.15 | 59.42 | 25.94 | 34.45 | 41.25 |
| CutMix (French et al., 2020) | **53.71** | 66.95 | 72.42 | 44.02 | 54.72 | 62.24 | 27.60 | 37.55 | 42.69 |
| ClassMix (Olsson et al., 2021) | 49.06 | 67.95 | 72.50 | 45.61 | 55.56 | 63.94 | 28.42 | 37.55 | 42.46 |
| ReCo + ClassMix | 53.31 | **69.81** | **72.75** | **49.86** | **57.69** | **65.04** | **29.65** | **39.14** | **44.55** |

Table 1: mean IoU validation performance for Pascal VOC, CityScapes and SUN RGB-D datasets. We report the mean over three independent runs for all methods.

To further justify the effectiveness of ReCo, we also include results on existing benchmarks, to compare with other semi-supervised methods in Table 2. Here, all baselines were re-implemented and reported in the PseudoSeg setting (Zou et al., 2021), where the labelled images are sampled from the original PASCAL dataset, with a total of 1.4k images. In both benchmarks, ReCo shows state-of-the-art performance, and specifically is able to reach PseudoSeg's performance, whilst *requiring only half the labelled data*. Additional results are further shown in Appendix C.

| Pascal VOC | 1/16 [92] | 1/8 [183] | 1/4 [366] | 1/2 [732] |
|---|---|---|---|---|
| AdvSemSeg (Hung et al., 2019) | 39.69 | 47.58 | 59.97 | 65.27 |
| Mean Teacher (Tarvainen & Valpola, 2017) | 48.70 | 55.81 | 63.01 | 69.16 |
| CCT (Ouali et al., 2020) | 33.10 | 47.60 | 58.80 | 62.10 |
| GCT (Ke et al., 2020) | 46.04 | 54.98 | 64.71 | 70.67 |
| VAT (Miyato et al., 2018) | 36.92 | 49.35 | 56.88 | 63.34 |
| CutMix (French et al., 2020) | 55.58 | 63.20 | 68.36 | 69.87 |
| PseudoSeg (Zou et al., 2021) | 57.60 | 65.50 | 69.14 | 72.41 |
| ReCo + ClassMix | **64.78** | **72.02** | **73.14** | **74.69** |

Table 2: mean IoU validation performance for Pascal VOC with data partition and training strategy proposed in PseudoSeg (Zou et al., 2021). The percentage and the number of labelled data used are listed in the first row.

In Fig. 5, we present qualitative results from the semi-supervised setup with the fewest labels: 20 labels for CityScapes and 50 labels SUN RGB-D datasets. The 60 labelled Pascal VOC is further shown in Appendix D. In Fig. 5, we can see the advantage of ReCo, where the edges and boundaries of small objects are clearly more pronounced such as in the `person` and `bicycle` classes in CityScapes, and the `lamp` and `pillow` classes in SUN RGB-D. Interestingly, we found that in SUN RGB-D, though all methods may confuse ambiguous class pairs such as `table` and `desk` or `window` and `curtain`, ReCo still produces consistently sharp and accurate object boundaries compared to the Supervised and ClassMix baselines where labels are noisy near object boundaries.

## 4.2 RESULTS ON PASCAL VOC AND CITYSCAPES (PARTIAL LABELS)

In the partial label setting, we evaluated on the CityScapes and Pascal VOC datasets. Table 3 compared ReCo to the two best semi-supervised baselines and a supervised baseline. Again, we see ReCo can improve performance in all cases when applied on top of ClassMix, with around $1 - 3\%$

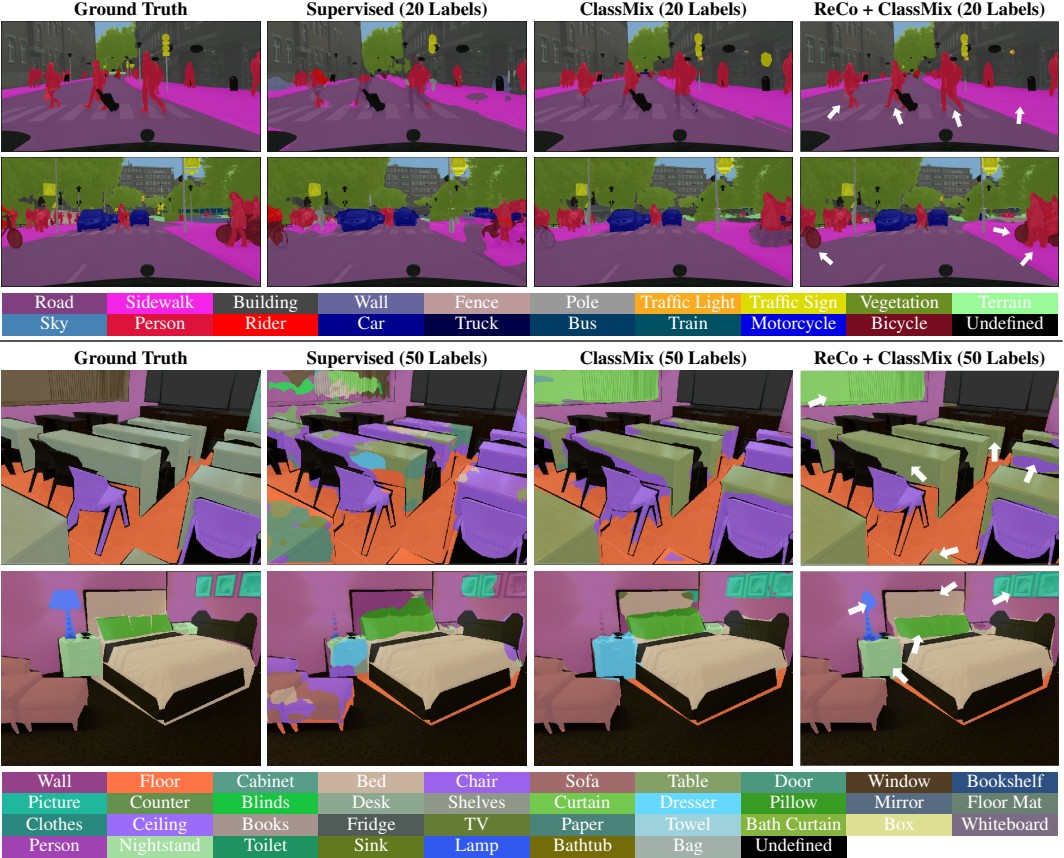

Figure 5: Visualisation of CityScapes and SUN RGB-D validation set trained on 20 and 50 labelled images respectively. Interesting regions are shown in white arrows.

| | Pascal VOC (Partial) | | | | | CityScapes (Partial) | | | |
|---|---|---|---|---|---|---|---|---|---|
| Method | 1 pixel | 1% labels | 5% labels | 25% labels | Method | 1 pixel | 1% labels | 5% labels | 25% labels |
| Supervised | 60.33 | 66.17 | 69.16 | 73.75 | Supervised | 44.08 | 52.89 | 56.65 | 63.43 |
| CutMix | 63.50 | 70.83 | 73.04 | 75.64 | CutMix | 46.91 | 54.90 | 59.69 | 65.61 |
| ClassMix | 63.69 | 71.04 | 72.90 | 75.79 | ClassMix | 47.42 | 56.68 | 60.96 | 66.46 |
| ReCo + ClassMix | **66.11** | **72.67** | **74.09** | **75.96** | ReCo + ClassMix | **49.66** | **58.97** | **62.32** | **66.92** |

Table 3: mean IoU validation performance for Pascal VOC and Cityscapes datasets trained on $1, 1\%, 5\%$ and $25\%$ labelled pixels per class per image. We report the mean over three independent runs for all methods.

absolute improvement. We observe less relative performance improvement than in the full label setting; very sparse ground-truth annotations could confuse ReCo, resulting in inaccurate supervision.

We show qualitative results on Pascal VOC dataset trained on 1 labelled pixel per class per image in Fig. 6. As in the full label setting, we see smoother and more accurate boundary predictions from ReCo. More visualisations from CityScapes are shown in Appendix E.

### 4.3 ABLATIVE ANALYSIS

Next we present an ablative analysis on 20 labelled CityScapes images to understand the behaviour of ReCo with respect to hyper-parameters. We use our default experimental setting from Section 4.1, using ReCo with ClassMix. Additional ablations are further shown in Appendix B.

**Number of Queries and Keys** We first evaluate the performance by varying the number of queries and keys used in ReCo framework, whilst fixing all other hyper-parameters. In Fig. 7a and 7b, we can observe that performance is better when sampling more queries and keys, but after a certain

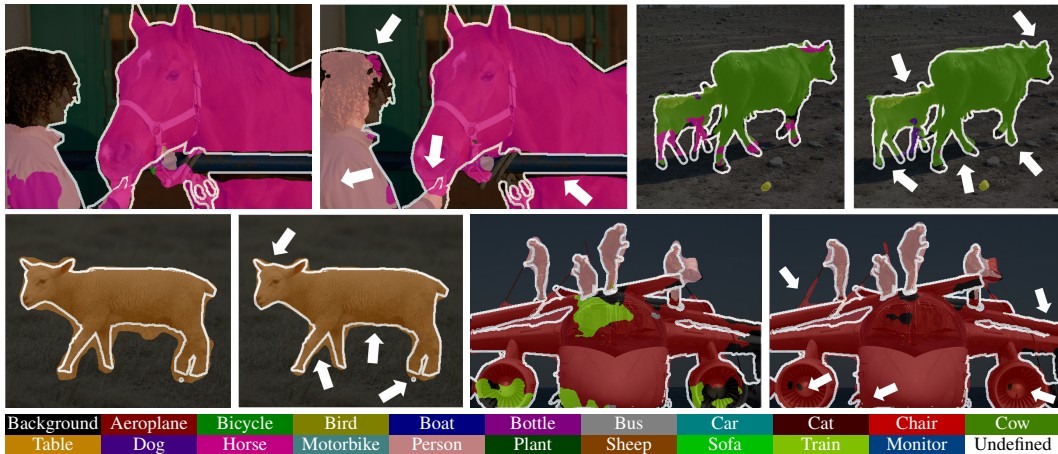

| Background | Aeroplane | Bicycle | Bird | Boat | Bottle | Bus | Car | Cat | Chair | Cow |
| Table | Dog | Horse | Motorbike | Person | Plant | Sheep | Sofa | Train | Monitor | Undefined |

Figure 6: Visualisation of Pascal VOC validation set with ClassMix (left) vs. with ReCo (right) trained on 1 labelled pixel per class per image. Interesting regions are shown in white arrows.

point, the improvements become marginal. Notably, even in our smallest option of having 32 queries per class in a mini-batch — consisting of less than 0.5% among all available pixel space — this can still improve performance by a non-trivial margin. Compared to a concurrent work (Zhang et al., 2021) which requires 10k queries and 40k keys in each training iteration, ReCo can be optimised with ×50 more efficiency in terms of memory footprint.

**Ratio of Unlabelled Data** We evaluate how ReCo can generalise across different levels of unlabelled data. In Fig. 7c, we show that by only training on 10% unlabelled data in the original setting, we can already surpass the ClassMix baseline. This shows that ReCo can achieve strong generalisation not only in label efficiency but also in data efficiency.

**Choice of Semi-Supervised Method** Finally, we show that ReCo is robust to the choice of different semi-supervised methods. In Fig. 7d, we can see that ReCo obtains a better performance from a variety choice of semi-supervised baselines.

**Effect of Active Sampling** In Fig. 7e, we see that randomly sampling queries and keys gives much less improvement compared to active sampling in our default setting. Particularly, hard query sampling has a dominant effect on generalisation: if we instead only sample from easy queries, ReCo only marginally improves on the baseline. This further verifies that most queries are redundant.

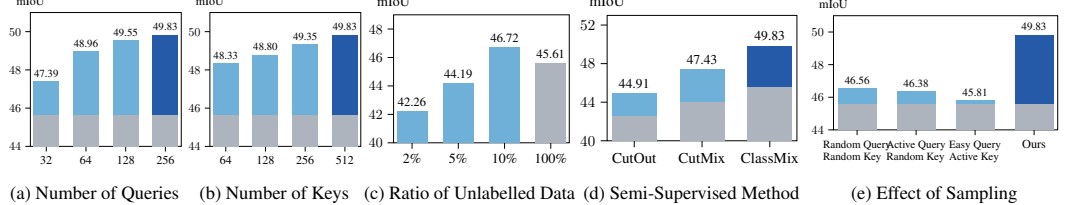

(a) Number of Queries  (b) Number of Keys  (c) Ratio of Unlabelled Data  (d) Semi-Supervised Method  (e) Effect of Sampling

Figure 7: mean IoU validation performance on 20 labelled CityScapes dataset based on different choices of hyper-parameters. Grey: ClassMix (if not labelled otherwise) in our default setting. Light Blue: ReCo + ClassMix (if not labelled otherwise) in a different hyper-parameter setting. Dark Blue: ReCo + ClassMix in our default setting.

## 5 VISUALISATIONS AND INTERPRETABILITY OF CLASS RELATIONSHIPS

In this section, we visualise the pair-wise semantic class relation graph defined in Eq. 3, additionally supported by a semantic class dendrogram using the off-the-shelf hierarchical clustering algorithm in SciPy (Virtanen et al., 2020) for better visualisation. The features for each semantic class used in both visualisations are averaged across all available pixel embeddings in each class from the validation set. In all visualisations, we compared features learned with ReCo built on top of supervised

learning compared to a standard supervised learning method trained on all data, representing the semantic class relationships of the full dataset.

Using the same definitions in Section 3.1, we first choose such pixel embedding to be the embedding $Z$ predicted from the encoder network $\phi$ in both supervised learning and with ReCo. We also show the visualisation for embedding $R$ which is the actual representation we used for ReCo loss and active sampling.

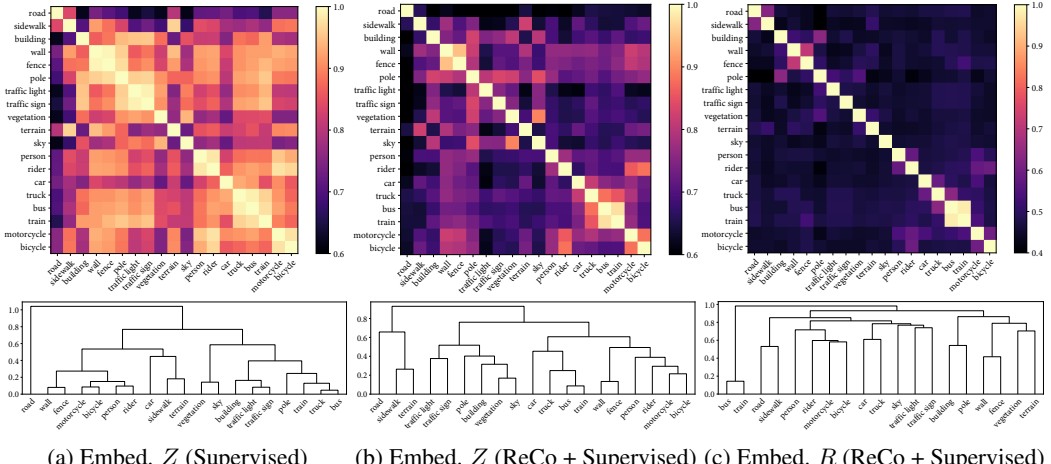

| (a) Embed. $Z$ (Supervised) | (b) Embed. $Z$ (ReCo + Supervised) | (c) Embed. $R$ (ReCo + Supervised) |

Figure 8: Visualisation of semantic class relation graph (top) and its corresponding semantic class dendrogram (bottom) on CityScapes dataset. Brighter colour represents closer (more confused) relationship. Best viewed in zoom.

In Fig. 8, we present the semantic class relationship and dendrogram for the CityScapes dataset by embedding $R$ and $Z$ with and without ReCo. We can clearly see that ReCo helps disentangle features compared to supervised learning where many pairs of semantic classes are similar. In addition, we find that the dendrogram generated by ReCo based on embedding $Z$ is more structured, showing a clear and interpretable semantic tree by grouping semantically similar classes together: for example, all large transportation classes car, truck, bus and train are under the same parent branch. In addition, we find that nearly all classes based on embedding $R$ are perfectly disentangled, except for bus and train, suggesting the CityScapes dataset might not have sufficient bus and train examples to learn a distinctive representation for these two classes.

The pair-wise relation graph helps us to understand the distribution of semantic classes in each dataset, and clarifies the pattern of incorrect predictions from the trained semantic network. We additionally provide a dendrogram based on embedding $R$ for the SUN RGB-D dataset, clearly showing ambiguous class pairs, such as night stand and dresser; table and desk; floor

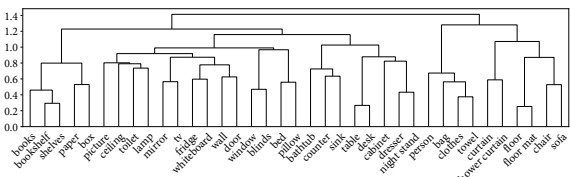

Figure 9: Visualisation of semantic class dendrogram based on embedding $R$ on SUN RGB-D dataset using ReCo + Supervised method. Best viewed in zoom.

and floormat, consistent with our results shown in Fig. 5. Complete visualisations of these semantic class relationships are shown in Appendix F.

## 6 CONCLUSION

In this work, we have presented ReCo, a new pixel-level contrastive framework with active sampling, designed specifically for semantic segmentation. ReCo can improve performance in semantic segmentation methods with minimal additional memory footprint. In particular, ReCo has shown its strongest effect in semi-supervised learning with very few labels, where we improved on the state-of-the-art by a large margin. In further work, we aim to design effective contrastive frameworks for video representation learning.

## REPRODUCIBILITY

All of the information for reproducibility is shown in Appendix A. Code is available at `https://github.com/lorenmt/reco`.

## ACKNOWLEDGEMENT

This work has been supported by Dyson Technology Ltd. We thank Zhe Lin for the initial discussion and Zhengyang Feng for his help on the evaluation metric design.

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

APPENDIX

## A    IMPLEMENTATION DETAILS

We trained all methods with SGD optimiser with learning rate $2.5 \times 10^{-3}$, momentum 0.9, and weight decay $5 \cdot 10^{-4}$. We adopted the polynomial annealing policy to schedule the learning rate, which is multiplied by $(1 - \frac{iter}{total\_iter})^{power}$ with $power = 0.9$, and trained for 40k iterations for all datasets. *Code is attached in the supplementary material.*

For CityScapes, we first downsampled all images in the dataset to half resolution $[512 \times 1024]$ prior to use. We extracted $[512 \times 512]$ random crops and used a batch size of 2 during training.

For Pascal VOC, we extracted $[321 \times 321]$ random crops, applied a random scale between $[0.5, 1.5]$, and used a batch size of 10 during training.

For SUN RGB-D, we first rescaled all images to $[384 \times 512]$ resolution, extracted $[321 \times 321]$ random crops, applied a random scale between $[0.5, 1.5]$, and used a batch size of 5. We additionally re-organised the original training and validation split in SUN RGB-D dataset from 5285 and 5050 to 9860 and 475 samples respectively, to increase the amount of training data which we think is more appropriate for semi-supervised task.

All datasets were additionally augmented with Gaussian blur, colour jittering, and random horizontal flip. The pre-processing for CityScapes and Pascal VOC are consistent with the prior work (Olsson et al., 2021). In Table 2, we extracted $[513 \times 513]$ random crops and applied a random scale between $[0.5, 2.0]$, following PseudoSeg's training setup (Zou et al., 2021).

In our ReCo framework, we sampled 256 query samples and 512 key samples and used temperature $\tau = 0.5$ for each mini-batch, which we found to work well in all datasets. The dimensionality for pixel-level representation was set to $m = 256$. The confidence thresholds were set to $\delta_w = 0.7$ and $\delta_s = 0.97$.

## B    ADDITIONAL ABLATIVE STUDIES

**ReCo Only Results**    Table 4 shows ReCo designed with and without data augmentation, trained on 20 and 50 labelled CityScapes dataset. We observe that using pure semi-supervised learning with additional unlabelled data will lead to worse performance compared to supervised learning without such labelled data. This shows data augmentation strategies designed for semi-supervised segmentation are the key component to make best use of the unlabelled data. Although the vanilla ReCo still performs better compared to standard semi-supervised learning, the active sampling of ReCo based on incorrect pseudo-labels leads to marginal improvement compared to a pure data augmentation method like ClassMix. Therefore, ReCo performs better as an auxiliary framework combined with a strong semi-supervised method.

| CityScapes | 20 Labels | 50 Labels |
|---|---|---|
| Supervised | 38.10 | 47.10 |
| Semi-Supervised | 28.59 | 43.74 |
| Semi-Supervised + ReCo | 29.16 | 46.96 |
| ClassMix | 45.61 | 55.56 |
| ClassMix + Reco | 49.86 | 57.69 |

Table 4: mean IoU validation performance for 20 and 50 labelled CityScapes data for supervised method (top) and semi-supervised methods (bottom).

**Compared to Feature Bank Methods**    We have experimented with ReCo with a stored feature bank framework similar to the design in the concurrent works (Alonso et al., 2021; Wang et al., 2021a). We found that just by replacing our batch-wise sampling method with a feature bank sampling method will achieve a similar performance (49.34 mIoU) compared to our original design (49.86 mIoU) on 20 labelled CityScapes, but with a slower training speed. This verifies our assumption that batch-wise sampling is an accurate approximation of class distribution over the entire dataset.

## C    RESULTS ON SEMI-SUPERVISED SEGMENTATION BENCHMARKS

Here, we present quantitative results for other semi-supervised semantic segmentation benchmarks in CityScapes and Pascal VOC datasets. Note that, this benchmark is much less challenging compared to our proposed benchmark in Section 4.1, evaluated with significantly less number of labelled images. Since some methods applied with different backbones and training strategies, we compared each result with respect to its performance gap compared to its corresponding fully supervised result, as shown in brackets, to ensure fairness following Feng et al. (2020).

In Table 5, we show results for ReCo applied on top of ClassMix, and trained with both DeepLabv2 (Chen et al., 2017) and DeepLabv3+ (Chen et al., 2018). We can observe that ReCo achieved the best performances in most cases in both datasets, showing its robustness to different backbone architectures and number of labelled training images.

| Pascal VOC | Backbone | 1/106 [100] | 1/50 [212] | 1/20 [529] | 1/8 [1323] | Full [10582] |
|---|---|---|---|---|---|---|
| AdvSemSeg (Hung et al., 2019) | DeepLabv2 | - | $57.20_{(17.70)}$ | $64.70_{(10.20)}$ | $69.50_{(5.40)}$ | 74.90 |
| S4GAN (Mittal et al., 2019) | DeepLabv2 | - | $63.30_{(12.30)}$ | $67.20_{(8.40)}$ | $71.40_{(4.20)}$ | 75.60 |
| CutMix (French et al., 2020) | DeepLabv2 | $53.79_{(18.71)}$ | $64.81_{(7.73)}$ | $66.48_{(6.06)}$ | $67.60_{(4.94)}$ | 72.54 |
| ClassMix (Olsson et al., 2021) | DeepLabv2 | $54.18_{(19.95)}$ | $66.15_{(7.98)}$ | $67.77_{(6.36)}$ | $71.00_{(3.13)}$ | 74.13 |
| CCT (Ouali et al., 2020) | PSPNet | - | - | - | $70.45_{(4.80)}$ | 75.25 |
| CAC (Lai et al., 2021) | PSPNet | - | - | - | $72.50_{(3.90)}$ | 76.40 |
| GCT (Ke et al., 2020) | DeepLabv2 | - | - | - | $70.57_{(3.49)}$ | 74.06 |
| DMT (Feng et al., 2020) | DeepLabv2 | $63.04_{(11.71)}$ | $67.15_{(7.60)}$ | $69.92_{(4.83)}$ | $72.70_{(2.05)}$ | 74.75 |
| ReCo + ClassMix | DeepLabv2 | $63.16_{(11.20)}$ | $66.41_{(7.95)}$ | $68.85_{(5.51)}$ | $71.00_{(3.36)}$ | 74.36 |
| ReCo + ClassMix | DeepLabv3+ | $63.60_{(14.15)}$ | $72.14_{(5.61)}$ | $73.66_{(4.09)}$ | $74.62_{(3.13)}$ | 77.75 |

| CityScapes | Backbone | 1/30 [100] | 1/8 [372] | 1/4 [744] | 1/2 [1488] | Full [2975] |
|---|---|---|---|---|---|---|
| AdvSemSeg (Hung et al., 2019) | DeepLabv2 | - | $58.80_{(7.60)}$ | $62.30_{(4.10)}$ | $65.70_{(0.70)}$ | 66.40 |
| S4GAN (Mittal et al., 2019) | DeepLabv2 | - | $59.30_{(6.50)}$ | $61.90_{(3.90)}$ | - | 65.80 |
| CutMix (French et al., 2020) | DeepLabv2 | $51.20_{(16.33)}$ | $60.34_{(7.19)}$ | $63.87_{(3.66)}$ | - | 67.53 |
| ClassMix (Olsson et al., 2021) | DeepLabv2 | $54.07_{(12.12)}$ | $61.35_{(4.84)}$ | $63.63_{(2.56)}$ | $66.29_{(-0.10)}$ | 66.19 |
| DMT (Feng et al., 2020) | DeepLabv2 | $54.81_{(13.36)}$ | $63.03_{(5.13)}$ | - | - | 68.16 |
| ECS* (Mendel et al., 2020) | DeepLabv3+ | - | $67.38_{(7.38)}$ | $70.70_{(4.06)}$ | $72.89_{(1.87)}$ | 74.76 |
| ReCo + ClassMix | DeepLabv2 | $56.53_{(12.07)}$ | $64.94_{(3.66)}$ | $67.53_{(1.07)}$ | $68.69_{(-0.09)}$ | 68.60 |
| ReCo + ClassMix | DeepLabv3+ | $60.28_{(10.20)}$ | $66.44_{(4.04)}$ | $68.50_{(1.98)}$ | $70.63_{(-0.15)}$ | 70.48 |

Table 5: mean IoU validation performance in semi-supervsed Pascal VOC and CityScapes datasets. We list the percentage along with the number of labelled images at the top row. The first and second best performances in each data partition setting are coloured in red and orange respectively. ∗ trained images in doubled resolution. All results were taken from the corresponding publications.

# D   VISUALISATION ON PASCAL VOC (TRAINED WITH 60 LABELLED IMAGES)

In the full label setting, the baselines Supervised and ClassMix are very prone to completely mis-classifying rare objects such as `boat, bottle` and `table`, while our method can predict these rare classes accurately.

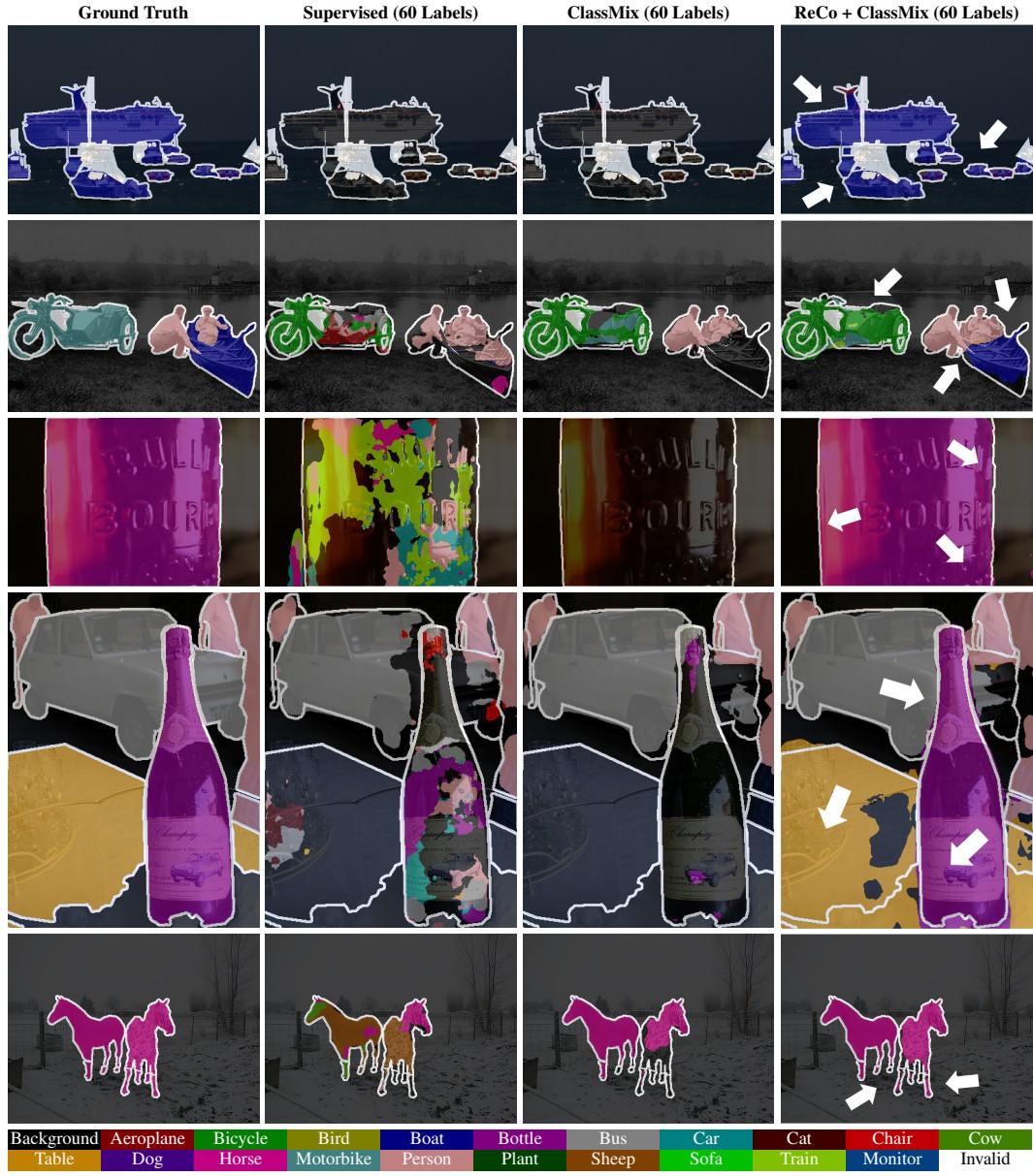

# E VISUALISATION ON CITYSCAPES (TRAINED WITH 1% LABELLED PIXEL)

In the partial label setting, the performance improvements are less pronounced compared to the full label setting in CityScapes dataset. The improvements typically come from the more accurate predictions in small object boundaries such as in `traffic light` and `traffic sign`. Learning semantics with partial labels with minimal boundary information remains an open research question and still has huge scope for improvements.

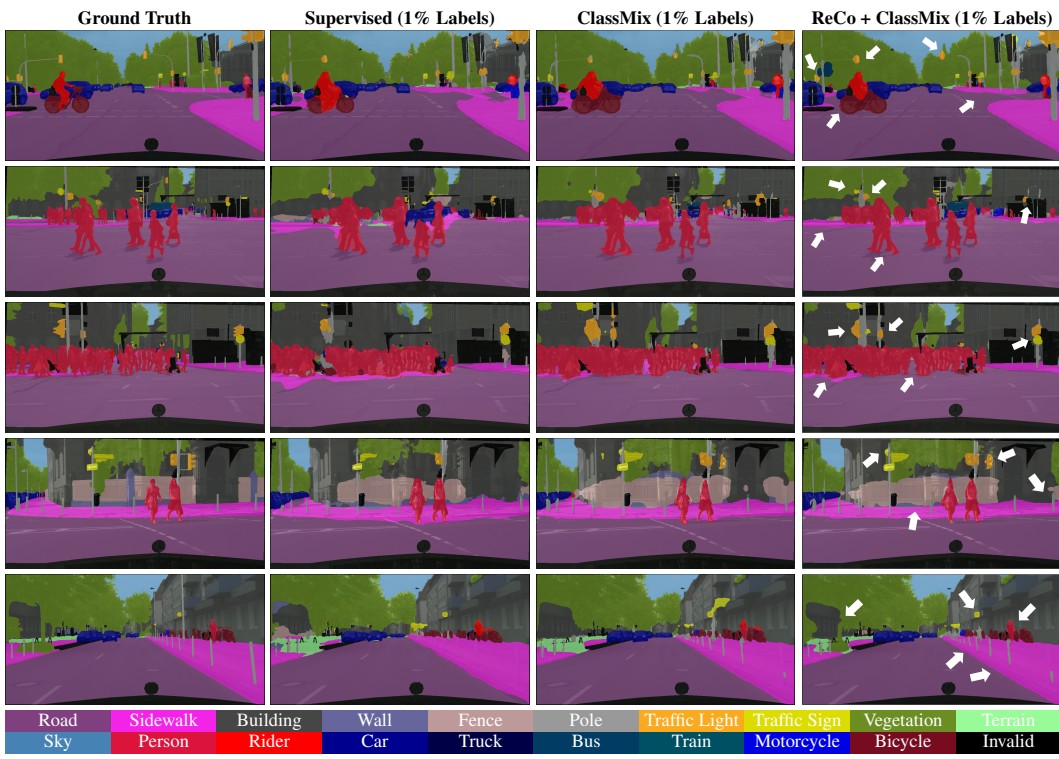

# F  VISUALISATION ON SEMANTIC CLASS RELATIONSHIP FROM PASCAL VOC (TOP) AND SUN RGB-D (BOTTOM)

We show features learned by ReCo are more disentangled compared to the Supervised baseline in all datasets, which helps the segmentation model to learn a better decision boundary. Brighter colour represents closer (more confused) relationship. Best viewed in zoom.

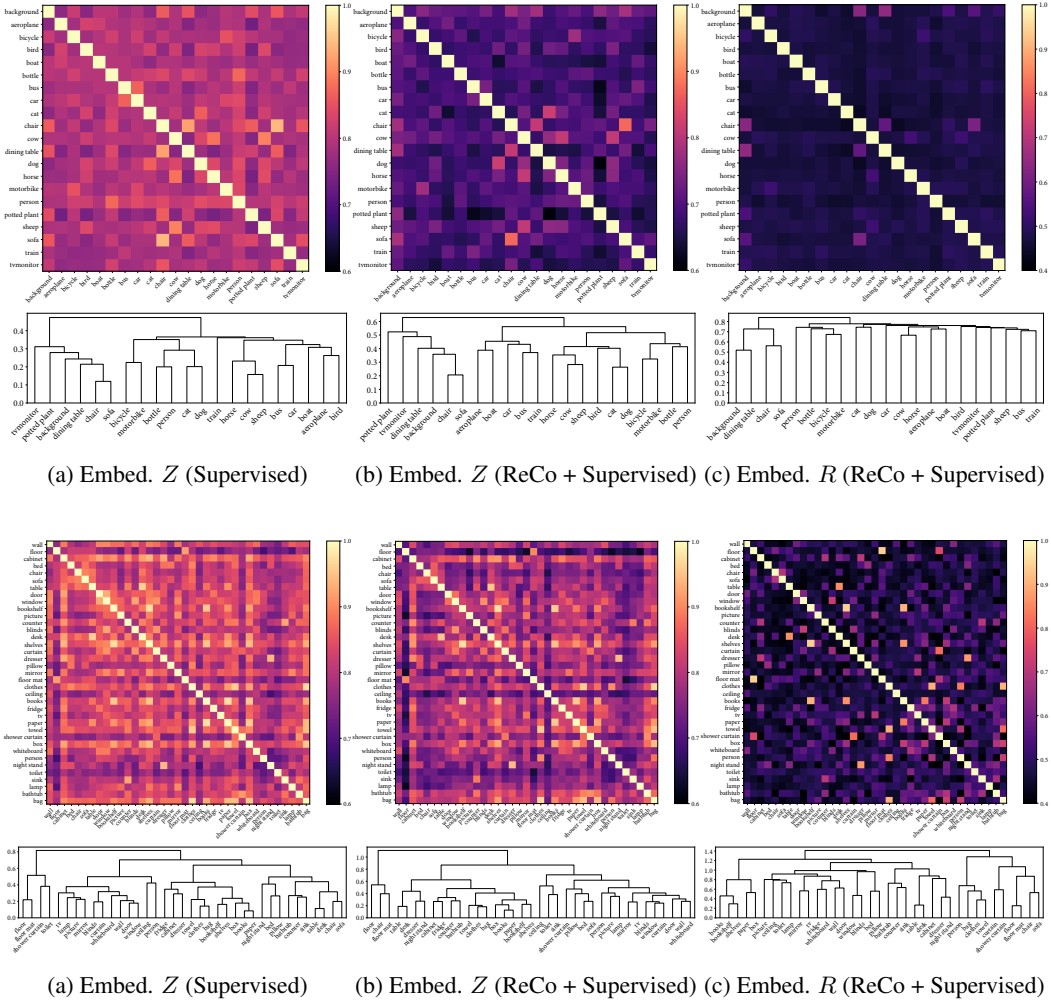

(a) Embed. $Z$ (Supervised)  (b) Embed. $Z$ (ReCo + Supervised)  (c) Embed. $R$ (ReCo + Supervised)

(a) Embed. $Z$ (Supervised)  (b) Embed. $Z$ (ReCo + Supervised)  (c) Embed. $R$ (ReCo + Supervised)

