# OpenReview forum: "Bootstrapping Semantic Segmentation with Regional Contrast"
_ICLR.cc/2022/Conference — ICLR 2022 Poster_

### Official Review · Reviewer_qBDp · 2021-10-25

**Correctness:** 4
**Technical Novelty And Significance:** 2
**Empirical Novelty And Significance:** 3
**Recommendation:** 6
**Confidence:** 4

**Main Review:**

=== Strength ===
1. The proposed method is reasonable and well-motivated.

2. The paper is very well-written and easy to understand.

3. The experimental results and analysis presented by the authors align well with the authors' claims.

4. The proposed method showed state-of-the-art level performances in various settings and various datasets.


=== Weakness ===
1. Novelty.
- The contrastive approach proposed in this paper seems to have slightly modified the contrastive learning technique used in classification. In addition, the mean-teacher method has been considered by many existing semi-supervised learning methods.

2. Experiment.
- The authors provide only the results of ReCo+ClassMix, and do not provide the results of ReCO itself.
- Baselines in Table 1 seem weak. Recent methods [ref1, ref2, ref3] should be compared.
- The strong baseline [ref2] is missing. [ref2] produces 72.4 (1/16), 74.6 (1/8), and 76.3 (1/4) with ResNet-101 DeepLab v3+.

3. Experimental settings
- Partial label setting is also important, but the weak+semi-supervised setting seems more important in this field, because it provides more diverse and strong baselines. I recommend the authors to conduct a weak+semi supervised setting experiment and compare it with strong baselines [ref1, ref2, ref3, ref4].


[ref1] Ouali, Yassine, Céline Hudelot, and Myriam Tami. "Semi-supervised semantic segmentation with cross-consistency training." Proceedings of the IEEE/CVF Conference on Computer Vision and Pattern Recognition. 2020.

[ref2] Lai, Xin, et al. "Semi-Supervised Semantic Segmentation With Directional Context-Aware Consistency." Proceedings of the IEEE/CVF Conference on Computer Vision and Pattern Recognition. 2021.

[ref3] Zou, Yuliang, et al. "Pseudoseg: Designing pseudo labels for semantic segmentation."ICLR. 2021.

[ref4] Lee, Jungbeom, Eunji Kim, and Sungroh Yoon. "Anti-Adversarially Manipulated Attributions for Weakly and Semi-Supervised Semantic Segmentation." Proceedings of the IEEE/CVF Conference on Computer Vision and Pattern Recognition. 2021.

4. Minor

- For background query pixels, is a single mean vector sufficient as a positive key?

- In the active negative key sampling method, once a hard class is sampled, are negative key pixels randomly selected among the pixels of that class?

**Summary Of The Paper:**

This paper proposes ReCo, a regional contrastive learning method for semi-supervised semantic segmentation. The query and key pixel sampling methods are proposed for efficient learning. The proposed method showed state-of-the-art level performances in various settings and various datasets.

**Summary Of The Review:**

Although this paper lacks novelty, I think that it will provide a good future direction for many researchers in this field, if it is verified that the proposed method generally works in more settings. If my concerns related to experiments are resolved, I would consider upgrading the score.

---

> ### Author Response · Authors · 2021-11-19
> **Response for R4**
>
> We thank R4 for the constructive comments and we hope to clarify your understanding in the following response.
>
> **Provide the results for ReCo itself**
>
> We agree that this would be an interesting ablation to study. We will include ReCo-only results in the new version of this paper. But please note that, because ReCo is designed as an *auxiliary* framework, it works best with other data augmentation methods designed for semi-supervised segmentation.  **(Proposed Change C)**
>
> **Comparing with [ref1,2,3]**
>
> We believe that R4 has overlooked our results in Table 2, which **already** included results for [ref1] and [ref3]. Also please note that we outperformed [ref3] whilst requiring only *half of the labelled data*, which we have highlighted in the paper.
>
> **Strong baseline of [ref2]**
>
> R4 mentioned that [ref2] produces 72.4 (1/16), 74.6 (1/8), and 76.3 (1/4) with ResNet-101 DeepLab v3+, which we agree. However, in [ref2], the labelled data portion 1/16, 1/8, 1/4 were sampled from the standard VOC12 **along with** its augmented set, which includes with total 10582 images. This is a different setting applied in Table 2, which represents the data portion in the standard VOC12 data only **without** the augmented set, which includes with total 1449 images. But, we have still already included the results in the [ref2] setting additionally in Appendix B, showing 1/50, 1/20, 1/8 with the same ResNet-101 DeepLab v3+ backbone will achieve 72.14, 73.66 and 74.62 respectively. Notice that, *we achieved [ref2]'s  1/16 data performance with only 1/50 data.* We will clarify this in the next version of the paper. **(Proposed Change D)**
>
> **Single mean vector sufficient for background class.**
>
> We agree that background class is complicated, consisting of more information than other classes. However, since we make *no assumption* on the design of dataset, we should not modify ReCo based on any semantic class or dataset.
>
> **Negative key pixels randomly selected.**
>
> The negative keys were sampled based on the learned distribution (Eq. 3), not based on a *single* hard class. But yes, these negative keys were sampled randomly from all features. Since only the hard queries would do the gradient update, we then encourage the hard queries pulling from *all* features from the confusing classes.
>
> **Proposed Changes: R4**
>
> Add ablation with only ReCo standalone method. **(Proposed Change C)** Clarify the experimental setting for Table 2. Add [ref2] results in Appendix B. **(Proposed Change D)**

---

> > ### Author Response · Authors · 2021-11-30
> > **Updated response for R4**
> >
> > Dear R4,
> >
> > Since we are closing to the end date of the open discussion, we would like to kindly remind you to read our rebuttal above which hopefully should resolve all your questions and comments in the reviews.
> >
> > We would love to hear your feedback and also love to answer if you have additional questions.

---

> > > ### Comment · Reviewer_qBDp · 2021-12-01
> > > **Final rating**
> > >
> > > Dear authors,
> > >
> > > I appreciate the rebuttal, which carefully addresses the comments of reviewers. The authors clearly pointed out my misunderstanding about the use of the augmented set of VOC in [ref2], thus I would upgrade my rating.
> > >
> > > However, my rating is "6: marginally above the acceptance threshold", not accept or higher, since I still think the novelty is limited considering the high standard of ICLR.

---

### Official Review · Reviewer_uAhS · 2021-10-31

**Correctness:** 3
**Technical Novelty And Significance:** 3
**Empirical Novelty And Significance:** 3
**Recommendation:** 6
**Confidence:** 4

**Main Review:**

Strengths:
* The proposed method can achieve high-quality semantic segmentation results with only 5 examples of each semantic class.
* The presentation and organization are clear and easy to follow.
* The figures in the paper look nice.
* Would be definitely accepted if it's presented six month ago.

Weakness:
* The similar idea have been presented in existing publications [1, 2] (appeared on arXiv before May 2021). [1] is missing in the relate work.

[1] Semi-Supervised Semantic Segmentation with Pixel-Level Contrastive Learning from a Class-wise Memory Bank. Alonso et al. ICCV 2021.

[2] Exploring cross-image pixel contrast for semantic segmentation. Wang et al. ICCV 2021

No considering the later one:

[3] Looking beyond single images for contrastive semantic segmentation. Zhang et al. NeurIPS 2021.

**Summary Of The Paper:**

The paper proposed to implement  the contrastive learning loss into the semi-supervised semantic segmentation framework. The method can achieve high quality semantic segmentation results by using only a few labeled samples (e.g 5 examples of each semantic class).

**Summary Of The Review:**

The contrastive learning for semi-supervised semantic segmentation has been presented in [1].
The differences/novelties compared to [1] are not significant enough for a new publication.

- Post-rebuttal:
I slightly improve my rating score from 3 to 5 (marginally below the acceptance threshold).
ICCV paper [1] is released on Arxiv in May, accepted in August and officially published in October.
Paper [1] must be addressed and compared in the paper. The author must make a clear statement on the differences and improvements (compared with [1]).

- For the rebuttal: "[1] and [2] both apply contrastive learning with stored feature banks, but we sample features on-the-fly".
The momentum memory bank has been shown effective in [MoCo, 1,2]. It boosts the scope for selecting the positive & negative pairs. "Sample features on-the-fly" can't be argued as an advantage over existing work without supporting experiments. Intuitively, this limits the selection of the contrastive pairs into the current mini-batch and would lead to worse accuracy. Experiments must be done to support this argument.

- After further discussion,
I would like to raise the rating score to 6 (marginally above the acceptance threshold).

---

> ### Author Response · Authors · 2021-11-19
> **Response for R3**
>
> We appreciate R3's help in bringing other concurrent works to our attention, and in acknowledging the qualitative results in our paper. R3 is mainly concerned with concurrent works published at ICCV and NeurIPS, but we would like to highlight that this **violates** ICLR's review policy.
>
> First of all, ICLR's official policy is that comparisons should **NOT** be requested for papers which were published in a peer-reviewed proceedings **less than four months** before the ICLR submission deadline. Even considering just the list of accepted papers (and not even the official proceedings), both these ICCV and NeurIPS accepted papers were released **after** the ICLR submission deadline, let alone within four months of the deadline. Whilst these papers were on ArXiv at the time of the ICLR deadline, we cannot be expected to compare to non peer-reviewed papers, as per the official ICLR rules. Second, the full paper of [3] was only publicly released at the **very same date of writing this rebuttal (Nov. 18th)**. So we were unable to access the paper even during the majority of the rebuttal period.
>
> Finally, although we agree that the design of ReCo is motivated by pixel-level contrastive learning which is similar to these works, they are **not strictly speaking concurrent works** with the exact same idea, particularly in the overall framework and experiments design. [1] and [2] both apply contrastive learning with stored feature banks, but we sample features on-the-fly. [1] additionally requires class-specific attention modules to determine the quality of sampled features, and with an additional entropy minimisation loss to regularise pseudo-label generation, whilst our approach samples hard features simply based on prediction confidence and class-dependent distribution, which leads to a much simpler design. [2] applies contrastive learning just for supervised learning, which is not the focus of this work. [3] applies contrastive learning based on generated auxiliary label maps via K-mean clustering, which is also a completely different framework design compared to our work.
>
> **Proposed Changes R3:**
>
> Improve the concurrent work section. **(Proposed Change A)**

---

> > ### Author Response · Authors · 2021-11-29
> > **Updated response for R3**
> >
> > First of all, we would like to thank R3 for raising the rating.
> >
> > Second, we would like to further emphasise that we have already compared the stored feature bank approach as similar to [1, 2] in our updated draft. Sorry for not highlighting this in the original rebuttal. Please see **(Proposed Change B)** in the general comment section. (And we will provide more discussions in our final paper.)  The stored feature bank sampling method eventually achieved a similar performance compared to our method, whilst having a slower training speed. Also, please note that -- the fact that the ICCV papers were accepted in August **was to inform the paper authors, not for the general public**. Considering the accepted paper list was released after the ICLR submission deadline,  it was *impossible* for us to know whether this paper got accepted in ICCV before submission.
> >
> > Finally, we are aware of MoCo and agree that MoCo improves the diversity of negative keys in contrastive learning. However, please note that the design of MoCo is applied to the *global representation* of images, which is **NOT** the same for pixel-level contrastive learning which is applied to the *dense representation* of pixels. We argue that the batch-wise active sampling proposed in the paper is more efficient and accurate based on the following observations and experiments:
> >
> > -- **Representation Sampling Space.** Contrastive learning on global representation requires a large number of negative keys for accurate gradient updates, and thus requires a large batch size. As R3 mentioned, MoCo elegantly resolves this by using a stored feature bank, which we definitely agree with. However, the number of negative keys in each image is sufficiently large for contrastive learning on dense (pixel-level) representation. Therefore, we argue that in pixel-level contrastive learning, we want to focus on sampling the **right** representations rather than **more** representations, further supported in the observation below.
> >
> > -- **Not all queries and keys are equal.** We have shown in the ablation section that just by sampling easy queries and/or random keys will have marginal performance improvement. Particularly, hard query
> > sampling has a dominant effect on generalisation: if we instead only sample from easy queries, ReCo
> > has nearly no improvement on the baseline. This confirms that the most queries and keys are redundant, and sampling more random queries and keys will not lead to better performance.
> >
> > -- **Limited Selection of Contrastive Pairs.** We have also discussed this in the R2's comment. Comparing to sampling all classes stored in a feature bank, ReCo have the following advantages: i) It can automatically ignore non-confusing classes by design, e.g. an airplane is much less likely to be in the same picture as a dog. ii) The sampled features for contrastive learning are always from the same segmentation model, whilst computing global distribution will require storing features in segmentation models from different time steps, which slows down training speed. Further, we visualised the learned distribution via dentrogram, as shown in Section 5, Fig. 8 and 9, with a clear interpretable relationships consistent with human intuition. Again, we have the supported experiment in the updated paper draft showing by replacing batch-wise sampling into feature bank sampling will not improve performance.
> >
> > I hope this resolves R3's concerns.

---

### Official Review · Reviewer_cwn7 · 2021-11-02

**Correctness:** 4
**Technical Novelty And Significance:** 3
**Empirical Novelty And Significance:** 3
**Recommendation:** 8
**Confidence:** 5

**Main Review:**

* Pros
1) The active query and key sampling seems to be effective in optimizing the contrastive loss with little computational overhead.
2) The exhaustive comparative study is provided. The performance was validated by re-implementing all existing methods in the same backbone architectures and training strategies for a fair comparison.
3) The performance gain over the existing methods is impressive in the semi-supervised semantic segmentation task.

* Cons
1) The sampling idea is rather simple. While the query is sampled using the pair-wise similarity measure between classes, the hard key is simply selected by thresholding the prediction confidence.
a) The pair-wise similarity is computed within mini-batches, meaning this may be unable to reflect the global distribution over entire datasets when sampling the active query.
b) The hard key sample is also selected with the simple thresholding. The performance may heavily depend on how to setup the threshold, and more importantly it is not considered that the threshold may be varying with respect to classes.

2) The pseudo labels for unlabeled images are generated by simply applying the threshold. This simple idea poses a risk of still generating unreliable pseudo labels. It would be better to refer to recent approaches proposed in the self-training of unsupervised domain adaptation (UDA).


**Summary Of The Paper:**

This work presents a new semi-supervised learning framework based on a pixel-level contrastive learning with active query/key sampling. For a given query feature with a class c, the positive key sample is generated as a mean vector of all samples with the class c, while the negative key sample is sampled by considering the pair-wise relationship between classes in (3) such that more samples are from hard cases. The active query is adaptively sampled by considering rare queries in order to overcome the class imbalance in semantic segmentation as in (4). The proposed contrastive loss, call ReCo, is applied to semi-supervised semantic segmentation task with two cases: full labels and partial labels. Extensive results demonstrate the outstanding performance when using ReCo loss within existing semi-supervised learning methods.

**Summary Of The Review:**

The proposed query/key sampling may be useful for the semi-supervised semantic segmentation task, and the idea was verified using extensive experiments. However, the simple thresholding used for sampling negative key and generating pseudo labels may be often sensitive when deployed in real-world dataset.

* Post-rebuttal: The rebuttal letter has addressed some concerns in the original comments. The additional ablation study in the appendix supports the effectiveness of computing the pair-wise similarity in mini-batches. The argument on the the fixed threshold is also reasonable. Thus, I improve my rating score from '6: marginally above the acceptance threshold' to '8: accept, good paper'.

---

> ### Author Response · Authors · 2021-11-19
> **Response for R2**
>
> We thank R2 for the support of this work and the recognition of the baseline re-implementation.
>
> Here are some detailed responses to your comments.
>
> **Pair-wise similarity computed in mini-batches will not reflect the global distribution over entire datasets.**
>
> We respectively disagree on this claim. We argue that the similarity measured in the mini-batches would lead to a much simpler and faster implementation, and is an *accurate approximation* of global distribution of the entire dataset. Comparing to sampling all classes stored in a feature bank, we have the following advantages: i) It can automatically ignore non-confusing classes by design, e.g. an airplane is much less likely to be in the same picture as a dog. ii) The sampled features for contrastive learning are always from the *same* segmentation model, whilst computing global distribution will require storing features in segmentation models from *different time steps*. Further, we visualised the learned distribution via dentrogram, as shown in Section 5, Fig. 8 and 9, with a clear interpretable relationships consistent with human intuition. We will perform one more ablation by sampling features based on global distribution with a stored feature bank in the updated draft to further confirm this. **(Proposed Change B)**
>
> **Threshold may be varying with respect to classes.**
>
> We agree that a better design is to have class-dependent thresholds. However, the design of ReCo assumes no prior knowledge available for the distribution of classes, and thus we use the same set of hyper-parameters for all experiments. An alternative approach such as thresholding via per-class percentage is also not a suitable solution, otherwise, we would under-sample some rare and small-size classes (such as traffic lights) and over-sample some common and large-size classes (such as sky and road). Using the same threshold as the measure of prediction confidence works well in practice across different datasets.
>
> **Self-training of unsupervised domain adaptation (UDA).**
>
> In our experiments along with additional results in Appendix B, we have tried to cover all state-of-the-art methods in semi-supervised segmentation, available at the time of our ICLR paper submission. To the best of our knowledge, self-training methods of unsupervised domain adaptation often require much more labelled data compared to standard semi-supervised methods. All self-training methods we included in the second paragraph of the Introduction section were trained with two datasets with full data, from which one dataset was trained with labels and another dataset was trained without labels. This is a completely different assumption in a semi-supervised segmentation task, where only *very few* labelled data are available, such as 20 labelled images in CityScapes in our setting. We are happy to further discuss this or compare with other self-training techniques if R2 could list some references.
>
> **Proposed Changes R2**:
>
> Add another ablation of sampling features with global distribution. **(Proposed Change B)**

---

### Official Review · Reviewer_qZuF · 2021-11-03

**Correctness:** 4
**Technical Novelty And Significance:** 2
**Empirical Novelty And Significance:** 3
**Recommendation:** 8
**Confidence:** 3

**Main Review:**

**Strengths**

i) Very illustrative experiments.

One nice additional experiment is the plotting of dendograms over the class representation. These provide some evidence that the representations make sense, independent of the effect on end-to-end accuracy.

There are also very extensive ablation & hyperparameter studies. The effect of the "active sampling" method is particularly interesting.

ii) Submission includes well-documented code.

The README gives full setup instructions and some sample training commands. Code for necessary setup to reproduce results, such as dataset preprocessing, is provided. The code is commented to an okay extent, and the names make sense. It is fairly straightfoward to find missing details in the code and resolve any confusions.

Minor comment: Code organization could be improved, especially in having a file "module_list.py" with a fairly generic name that includes classes & functions with very disparate purposes.

iii) Strong accuracy results.

The authors set up an experimental baseline with the state of the art in data augmentation, and extensively use all the available similar "tricks" to improve accuracy. The ReCo loss successfully improves accuracy on top of these as well.

The baselines (on which the submission still improves) are solid overall: "most of the time surpassing, the performance reported in each publication." For example, the authors report 72.9 mIoU for ClassMix on 5% of labels in PASCAL VOC, vs 67.77 mIoU reported in Table 2 of Olsson et al. This is unsuprising because this submission uses DeepLabV3+ as the baseline network while Olsson et. al. use DeepLabV2. But it is good that the authors resolve this confound and report the best version of baseline methods for comparison.

**Weaknesses**

iv) No empirical comparisons to other representation-based methods.

Unless I missed it, I don't see experimental comparison to what seems to be the most closely-related works: those under the "Contrastive Learning" subsection. Even if the other works don't e.g. emphasize semi-supervised learning, I would guess at least a partial comparison could be made given that the submission does also support fully-supervised training in the code. (As it naturally would, given that the upper bound of the % of artificially-sampled labels to use is 100%.)

It's less than ideal to see a submission propose both a new benchmark (as described in the "Semi-Supervised Segmentation Benchmark Redesign" subsection) and a new method simultaneously, with the method not also evaluated on existing simpler benchmarks.

v) Lack of ablation on the "Mean Teacher" component.

The final method proposed in the submission includes a union of a lot of the techniques in the literature for semi-supervised segmentation. So it would take quite a lot of experiments, on top of the already-extensive ones that were done, to consider ablating everything. One relatively ablation that was elided was using the original network as for pseudo-labels.

**Minor Comments**

  * Weird grammar in "better performance from a variety choice of semi-supervised baselines" on page 8. I'm not entirely sure what the intended meaning is here.
  * which -> that above Figure 8
  * Instructions on setting up CityScapes dataset for experiments seems to assume that zip files add an enclosing directory:
    For example, with the basic `unzip` command on ubuntu, gtFine_trainvaltest.zip unpacks into gtFine/. The cityscapes_preprocess.py script, though, seems to assume it goes in gtFine_trainvaltest/gtFine/. This then also fails silently with empty train_label_list, causing failures later. Documenting the expected cityscapes directory layout in README.md is a common practice, I recommend it here. See for example:

https://github.com/google-research/deeplab2/blob/main/g3doc/setup/cityscapes.md

I also recommend some assertions on the input in cityscapes_preprocess.py to catch these kinds of errors.

  * A possibly slightly closer comparison for contrastive loss in semantic segmentation is "Contrastive Learning for Label-Efficient Semantic Segmentation" by Zhao et al. Like (Wang et al., 2021a) it is roughy concurrent work (also published in ICCV 2021. Zhao et. al. also considers semi-supervised learning (though not as much of a focus as in the submission).
  * Suggest repeating description from Figure 8 that "Brighter color represents closer... relationship" in Appendix E as well (assuming the figures there have the same meaning).

**Summary Of The Paper:**

The paper extends semantic segmentation models with a contrastive loss on "representations" at different locations in the image. A separate "representation head" is built from the intermediate features (between the "encoder" and "decoder") of the fully-convolutional segmentation network (DeepLabV3+). The "regional contrast" (ReCo) loss then encourages self-similarity between the representations of pixels belonging the same class and less similarity between representations for pixels of different classes. The submission investigates adding the ReCo loss both fully supervised and semi-supervised semantic segmentation training on Pascal VOC, CityScapes, and SUN RGB-D benchmarks.

**Summary Of The Review:**

This is an interesting approach, and the authors provide extensive high-quality experiments that cover both end-to-end accuracy and illustrating how the method works.

---

> ### Author Response · Authors · 2021-11-19
> **Response for R1**
>
> We appreciate R1's support of this work, particularly for the effort put into looking deep into our code, and re-confirming that our implementation of baselines is strong and fair. *We always want to perform reproducible and open machine learning research*. Since there is no general implementation available for these baselines under the same training strategy, we have tried our best effort to reproduce baselines with a fair and challenging benchmark. We will definitely further improve the documentation and organisation of our code after the paper acceptance.
>
> Here are some detailed responses to your other comments.
>
> **Comparison to other representation-based methods.**
>
> We agree that comparing with other contrastive learning methods in a simpler benchmark, e.g. a supervised learning setting, would be interesting. However, the focus of our work, as highlighted in the general comments section, is to perform semantic segmentation with *very few labels*. Besides, training in a supervised learning setting would additionally require advanced architecture design and a completely different training strategy than semi-supervised methods, which will deviate from the original focus of this work.
>
> In addition, the papers introduced under the "Contrastive Learning" section, including another related paper (Zhao et al ICCV) suggested by R1, are *pre-training* methods for self-supervised representation learning. The design is very different from our method, which is an auxiliary framework designed specifically for semi-supervised segmentation. We highlighted this difference in the second paragraph of the "Contrastive Learning" section. However, we still plan to compare with another simple representation learning method with a stored feature bank **(See Proposed Change B in R2's comment)**, and will include more related works in the updated draft **(See Proposed Change A in R3's comment)**.
>
> **A new benchmark and a new method.**
>
> As mentioned in the comment above, there are not many benchmarks evaluated with the same backbone and with the same training strategies available for semi-supervised segmentation. We therefore proposed a new benchmark in order to ensure a *fair comparison*. We also have provided results with existing benchmarks proposed from other works. For example, in Table 2, we evaluated ReCo in the PseudoSeg's setting, and we also provided additional results with more benchmarks listed in Appendix B.
>
> **Ablation on the Mean Teacher.**
>
> Mean Teacher framework is now the essential component to stabilise  pseudo-label prediction in modern semi-supervised methods (in both classification and segmentation tasks). It was used in **all** baseline methods, and thus is not a **unique design element** for ReCo. Therefore, we believe that performing ablations on Mean Teacher framework would not give us additional insight into ReCo.
>
>
> **Proposed Changes R1:**
>
> We will rewrite our concurrent work section to include other related papers up-to-date, and we will highlight the difference between these works. **(Proposed Change A)** We will also compare with one more representation learning-based method using stored feature bank in ablation section. **(Proposed Change B)**

---

> > ### Comment · Reviewer_qZuF · 2021-11-29
> > **Response to new results**
> >
> > The changes proposed in the authors' rebuttal all sound good. Additional ablations somewhat also resolve the "new benchmark *and* new method" concern, as the ablations validate the (ablated portions of the) method almost as well as comparing to other papers. I continue to recommend acceptance.

---

### Author Response · Authors · 2021-11-19
**General Comments**

We thank all of the reviewers for their time and effort in providing these helpful comments. For simplicity, we rename Reviewers qZuF, cwn7, uAhS, and qBDp to R1, R2, R3, and R4, respectively. For each reviewer, we provide a detailed, individual response to their review, including our proposed changes to the paper based on their review. The rest of this comment here, summaries all of the changes which will are making to the paper.

Due to the limited time in the paper discussion period, it is unlikely that we will be able to finish all proposed experiments before the draft updating deadline (Nov. 22nd). However, we will continue to provide additional results in the comments section after this deadline, and we will include all proposed changes in the final version of this paper.

To summarise, we will make the following changes to the paper:

**Improve Related work. (R1, R3) (Proposed Change A)**

We will discuss and include additional related works published at ICCV and NeurIPS, as suggested by R1 and R3. However, we want to emphasise that, according to the Reviewer Guidelines of ICLR:

> We consider papers contemporaneous if they are published (available in online proceedings) within the last four months. That means, since our full paper deadline is October 5, if a paper was published (i.e., at a peer-reviewed venue) on or after June 5, 2021, authors are not required to compare their own work to that paper. '

Please note that the accepted paper lists for both ICCV and NeurIPS were released **after** the ICLR submission deadline (Oct. 7th for ICCV; Nov. 17th for NeurIPS), and so we would like to highlight that we are not required to experimentally compare to these works. For example, one NeurIPS paper (Zhang et al; [3]) suggested by R3 was **not even publicly released** until near the end of the rebuttal period. We are very happy to include these works in the related work section of our updated paper, but we hope reviewers understand that we cannot experimentally compare to these works.

Furthermore, we want to highlight that the focus of this work is to design ReCo, as a simple framework, for a very challenging setup in semi-supervised segmentation: training with extremely sparse labels, e.g. with only 5 samples per semantic class or 1 labelled pixel per class per image. And additionally, we aim to understand the relationships and distribution of these semantic classes, with the learned representation by ReCo. This is a very different research problem from these related works, which either trained with hundreds or thousands of labelled images [1,3,4], or were designed with additional regularisation losses and larger memory [2].

But we will include all of these papers and highlight the differences in the related work section.

[1] Semi-Supervised Semantic Segmentation with Pixel-Level Contrastive Learning from a Class-wise Memory Bank. Alonso et al. ICCV 2021.

[2] Exploring cross-image pixel contrast for semantic segmentation. Wang et al. ICCV 2021

[3] Looking beyond single images for contrastive semantic segmentation. Zhang et al. NeurIPS 2021.

[4] Contrastive Learning for Label-Efficient Semantic Segmentation. Zhao et al. ICCV 2021.

**Additional Ablation. R1, R2, R4 (Proposed Change B, C)**

We are going to add two additional ablation experiments. i)  to replace the current design of batch-wise online feature sampling with stored feature-bank sampling **(Proposed Change B)**, and ii) to train vanilla ReCo without data augmentation strategies **(Proposed Change C)**.  The first experiment shows whether batch-wise sampling is an accurate approximation of semantic class distribution learned for the entire dataset. And the second experiment verifies whether the data augmentation strategy is the key component to sample high-quality features.

**Clarification of Labelled Data Selection in Table 2. R4 (Proposed Change D)**

We will also improve and clarify the labelled data selection strategy used for Table 2.

---

> ### Author Response · Authors · 2021-11-22
> **Updated Paper Draft**
>
> We have now updated our paper with these proposed changes. We marked the changes in the color purple.

---

### Decision · Program_Chairs · 2022-01-20

**Decision:**

Accept (Poster)

**Comment:**

An interesting paper, with non-trivial results. The reviewers all agree that the paper is above bar (with two of them indicating strong vote for acceptance). The simplicity of the proposed approach (noted by some of the reviewers) is in my view a positive. Overall, a worthy contribution.